# General alkyl fluoride functionalization via short-lived carbocation-organozincate ion pairs

D. Lucas Kane [1], Bryan C. Figula[1], Kaluvu Balaraman [1], Jeffery A. Bertke [1] & Christian Wolf [1] ✉

Fluorinated organic compounds are frequently used across the chemical and life sciences. Although a large, structurally diverse pool of alkyl fluorides is nowadays available, synthetic applications trail behind the widely accepted utility of other halides. We envisioned that $C(sp^2)$-$C(sp^3)$ cross-coupling reactions of alkyl fluorides with fluorophilic organozinc compounds should be possible through a heterolytic mechanism that involves short-lived ion pairs and uses the stability of the Zn-F bond as the thermodynamic driving force. This would be mechanistically different from previously reported radical reactions and overcome long-standing limitations of organometallic cross-coupling methodology, including competing β-hydride elimination, homo-dimerization and hydrodefluorination. Here, we show a practical $C_{sp3}$-F bond functionalization method that expands the currently restricted synthetic space of unactivated primary, secondary and tertiary $C(sp^3)$-F bonds but also uses benzylic, propargylic and acyl fluorides. Many functional groups and sterically demanding substrates are tolerated, which allows practical carbon-carbon bond formation and late-stage functionalization.

The development of methods that allow selective carbon-carbon bond formation with high functional group tolerance is one of the most fundamental tasks in synthetic chemistry, especially if it generates opportunities with a wide range of readily available compounds and achieves currently not possible transformations including late-stage functionalization. During recent years, alkyl fluorides have become very popular building blocks primarily because of the importance of fluorinated organic compounds in the pharmaceutical and agro-chemical industries. The high demand is matched by a rapidly expanding pool of commercially available alkyl fluorides that now await extensive utilization in chemical synthesis. Significant progress has been made with the introduction of methods that apply this compound pool in reactions maintaining the $C(sp^3)$-F bond which is widely considered chemically inert due to its low polarizability and high bond dissociation energy exceeding the thermodynamic stability of all other single bonds that carbon can form. With the exception of elimination and hydrodefluorination reactions that reduce C-F to C-H bonds[1–9], generally applicable $C(sp^3)$-F bond functionalization has remained a major obstacle and is trailing far behind the striking synthetic utility of other alkyl halides which are routinely used for C-C bond construction. Reactions with monofluorinated compounds are challenging unless the C–F bond is activated, for example in allylic or benzylic positions[10–16].

Developments with unactivated alkyl fluorides are relatively scarce and few reports of chemodivergent C-C bond formation[17], defluorinative degradation[18], substitution and Friedel-Crafts reactions[19,20] have emerged. However, a narrow substrate scope, low functional group tolerance and competing side reactions are persisting problems, especially in cross-coupling chemistry with aryl Grignard reagents[21–27]. This has been only partially addressed with transition metal catalyzed niche applications as yield-limiting side reactions continue to interfere and often prevail[28–32]. As a result, currently possible Kumada-Corriu and Suzuki reactions utilize a relatively small selection of alkyl fluorides and organometallic

[1]Georgetown University, Chemistry Department, Washington, DC 20057, USA. ✉e-mail: cw27@georgetown.edu

reagents that may exhibit few functional groups, if any. In addition, sterically hindered aryl Grignards or boronic acids are not compatible and highly substituted C-C bonds cannot be formed. The synthetic value of alkyl fluorides thus remains limited by moderate yields, a narrow application scope, low functional group compatibility, and the absence of late-stage functionalization examples.

We expected that the introduction of a synthetic method that allows generally applicable carbon-carbon bond formation with alkyl fluorides and organozinc compounds would face several hurdles and likely require a mechanistic pathway that is different from currently known radical reactions with alkyl chlorides, bromides or

other electrophiles[33–35]. Since homolytic cleavage would be unlikely to overcome the chemical inertness and unmatched bond dissociation energy of the C(sp$^3$)-F functionality, we favored an ionic alkyl fluoride functionalization pathway that exploits the fluorophilicity of zinc(II) Lewis acids and proceeds via aryl(fluoro)zincate species activated for aryl transfer and C-C bond formation. Unfortunately, we found that there is neither a literature precedent of C(sp$^3$)-F bond cleavage with an diorganozinc compound nor structural information on arylzinc fluoride motifs that we envisioned would be formed in a heterolytic mechanism. In fact, the nonexistence of synthetic applications and the dearth of structural

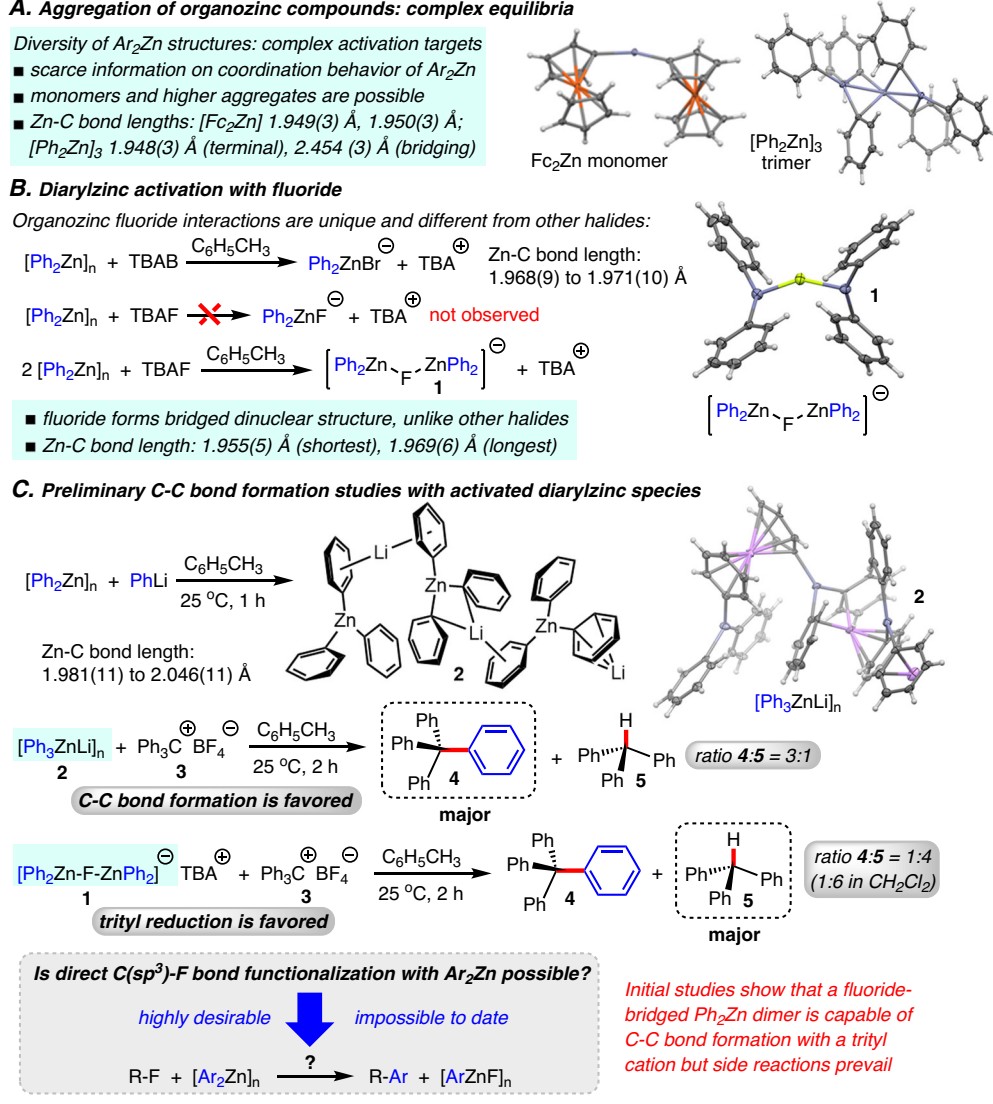

**Fig. 1 | Mechanistic investigation into the structure, reactivity and activation of diarylzinc compounds, arylzinc fluoride aggregates and higher organozincates. A** Crystallographic analysis reveals structural complexity and diversity of arylzinc compounds. **B** Fluoride bridges Ph$_2$Zn unlike other halides, indicating a higher propensity toward aggregation and different reactivity. **C** Preliminary development of aryl transfer chemistry with Ph$_3$ZnLi and fluoride-activated Ph$_2$Zn showing proof-of-concept among multiple challenges and possible pitfalls.

**A. Initial reaction screening with a tertiary fluorooxindole**

| Solvent | T (°C) | Time (h) | Conversion to 7 (%) |
|---|---|---|---|
| PhCF₃ | 90 | 18 | 80 |
| cyclohexane | 90 | 18 | 55 |
| DCE | 90 | 18 | 59 |
| PhCF₃ | 25 | 18 | 0 |
| PhCF₃ | 40 | 18 | 64 |
| **PhCF₃** | **60** | **18** | **80** |
| PhCF₃ | 60 | 6 | 74 |

Optimized conditions are shown in bold.

**B. Primary and secondaryl alkyl fluorides**

| R-F | T (°C) | Conversion to 10 (%) | R-F | T (ºC) | Conversion to 10 & 11 (%) |
|---|---|---|---|---|---|
| 8 | 60 | 6 | 8 | 90 | 62ᵉ |
| 8 | 90 | 17 | **8** | **70** | **73ᵉ** |
| 8 | 120 | 36 | 9 | 90 | 0ᶠ |
| 8 | 90ᵃ | 12 | 9 | 120 | 22ᶠ |
| 8 | 90ᵇ | 9 | 9 | 90 | 10ᵈ |
| 8 | 90 | 27ᶜ | 9 | **120** | **72ᶜ,ᵍ** |
| 8 | 90 | 68ᵈ | 9 | 120 | 72ᶜ,ᵍ,ʰ |
| 8 | 50 | 41ᵈ | 9 | 120 | 71ᶜ,ᵍ,ⁱ |

Optimized conditions are shown in bold. ᵃMicrowave (100 W, 90 ˚C, 1 h), ᵇ(300 W, 90 ˚C, 5 h), ᶜ96 h, ᵈ2 equiv. LiI (the reaction with **9** gave 80% octyl iodide), ᵉneat, ᶠ0.1 mL solvent, ᵍ0.02 mL, ʰ2 equiv of [Ph₂Zn]ₙ, ⁱ4 equiv of [Ph₂Zn]ₙ

**C. Reaction scope with Ph₂Zn**

- wide scope: 1º, 2º, 3º, unactivated, benzylic, propargylic, acyl fluorides
- selective C-F bond activation, high functional group tolerance
- access to crowded tetrasubstituted carbon scaffolds

**11**, 64%

**14**, 59%

**15**, 64%

**16**, 66%

**10**, 64%

**17**, 58%

**18**, 68%

**19**, 75%

**20**, 72%

**21**, 75%

**22**, 70%

**23**, 91%

**24**, 81%

**25**, 60%

**4**, 73%

**26**, 91%

**7**, 68%

**27**, 70%

**28**, 63%
(from acyl fluoride)

**29**, 77%
(from acyl fluoride)

**30**, 80%
(from acyl fluoride)

**Fig. 2 | Development of cross-coupling reactions between alkyl fluorides and diphenylzinc. A** Initial reaction screening with *N*-phenyl 3-fluorooxindole. **B** Optimization studies with 1-phenyl-3-fluorobutane and 1-fluorooctane representing unactivated secondary and primary alkyl fluorides. Conversions to the desired products were determined by GC-MS and NMR spectroscopic analysis.

**C** Scope and functional group tolerance of the C(sp²)-C(sp³) bond formation with primary, secondary, tertiary alkyl fluorides and acyl fluorides. All yields are based on isolated products. See Sections 4 and 5 in the Supplementary Information for details. equiv. equivalents.

knowledge of organozinc fluoride compounds are in stark contrast with the widespread occurrence of organozinc chlorides, bromides and iodides which are popular starting materials for organic synthesis and frequently encountered reaction intermediates[36–39]. To date, three reports on alkylzinc fluorides have appeared[40–42] but there is no crystallographic information on arylzinc derivatives and

their role in organic reactions including C-F bond functionalization chemistry is unexplored.

Here, we describe a broadly useful Csp3-F bond functionalization strategy that overcomes long-standing limitations of many organometallic C(sp²)-C(sp³) cross-coupling methods, including predominant side reactions such as β-hydride elimination, homodimerization, alkyl

halide reduction and protodemetalation, and we demonstrate the mechanistic underpinnings and scope of a practical solution to these problems. This is achieved with a reciprocal activation concept that exploits the high fluorophilicity of diarylzinc compounds in non-coordinating solvents and fast aryl transfer kinetics of short-lived carbocation-fluorozincate ion pair intermediates while the stability of the Zn-F bond serves as thermodynamic sink. We demonstrate that this chemistry is compatible with a wide variety of 1°, 2° and 3° alkyl fluorides, tolerant of many functional groups, capable of producing highly congested scaffolds, applicable to late-stage functionalization of pharmaceutically and agrochemically relevant scaffolds, and mechanistically distinct from previously reported methods.

## Results

### Investigation of the structure and reactivity of diarylzinc compounds

At the onset of this study, we realized that C-F bond functionalization with diarylzinc reagents in noncoordinating solvents is likely to involve complicated equilibria. For >30 years, diphenylzinc has been assumed to exist as a dimer in the solid state[43]. Careful single crystal growth and X-ray analysis by our laboratory confirmed the dimeric structure but also revealed that its propensity for aggregation is higher than commonly believed. We identified a trimeric aggregate with both terminal and unsymmetrically bridging phenyls exhibiting significantly elongated Zn-C bonds of up to 2.454(3) Å (Fig. 1A and Supplementary Fig. 1). By contrast, we found that diferrocenylzinc, $Fc_2Zn$, is a monomeric structure with short terminal Zn-C bonds ranging from 1.949(3) to 1.950(3) Å, indicating a previously unnoticed diversity of organozinc compounds (Supplementary Fig. 2). We then continued with investigating the possibility of diarylzinc activation with fluoride. In accordance with the few literature reports on organometallic fluoride complexes mentions above, we observed that organozinc fluoride chemistry does not follow the general trends of other halides. In fact, bromide coordination to $Ph_2Zn$ is known to generate a monomeric zincate[44], whereas the dinuclear structure 1 with a bridging fluoride and weakly elongated Zn-C bonds (1.955(5) to 1.969(6) Å) was obtained when we treated diphenylzinc with tetrabutylammonium fluoride (TBAF) in toluene (Fig. 1B and Supplementary Fig. 3). Despite the apparently weak activation in the $[Ph_2Zn(\mu_2\text{-}F)ZnPh_2]^-$ motif, we decided to employ it in preliminary C-C bond formation studies (Fig. 1C). For comparison, $[Ph_3ZnLi]_n$, 2, was prepared and its structure was determined by X-ray analysis (Supplementary Fig. 4 and 5). As expected, the reaction between this activated zincate and trityl tetrafluoroborate, 3, gave tetraphenylmethane, 4, as the major product in 75% yield. Unfortunately, the desired compound 4 was formed in only small amounts while predominant trityl reduction toward 5 and formation of other by-products were observed when 1 was used under the same conditions (Supplementary Figs. 6–9). These initial results showed proof-of-concept but also revealed major challenges. In addition, it was unknown whether diarylzinc compounds would be sufficiently fluorophilic to enable heterolytic fluoride abstraction. The organozinc fluorophilicity was expected to be further reduced by the presence of amino, alkoxy, carbonyl and other coordinating groups generally encountered in multifunctional substrates which would severely limit the scope of $C(sp^3)$-F bond functionalization. These potential pitfalls were matched by the scarce information on the structure, stability and reactivity of organozinc fluorides that appeared to form aggregates causing sluggish reactions and problems with insoluble intermediates. Finally, the weak aryl transfer capacity identified by the test reaction between 1 and 3 was a major concern because it would favor notorious side reactions, such as rearrangements via hydride migration, HF elimination and hydrodefluorination, that are generally fast and therefore likely to outcompete the desired C-C bond formation.

### Cross-coupling method development and reaction scope with diphenylzinc

These considerations and our initial mechanistic insights into aryl(fluoro)zincate reactivity and coordination chemistry provided important guidance for the following reaction development which was carried out with a set of three representative alkyl fluorides (Fig. 2 and Supplementary Tables 1–3). For this purpose, we chose 3-fluorooxindole, 6 a synthetically attractive tertiary fluoride with an expected high propensity for HF elimination, and the unactivated primary and secondary substrates 8 and 9 exhibiting particularly challenging $C(sp^3)$-F bonds. We were pleased to find that the conversion of 6 to the desired coupling product 7 occurs smoothly with stoichiometric amounts of diphenylzinc at ambient temperatures in several organic solvents (Fig. 2A). In fact, 7 was formed by gentle warming of the reaction mixture to 40 °C in trifluorotoluene, a practical and inexpensive solvent choice, but the reaction was still incomplete after 18 h. Increasing the temperature to 60 °C allowed complete conversion within the same time and we were able to effectively control competing HF elimination and hydrodefluorination pathways which gave -10% of the corresponding oxindole by-products. Since $C(sp^2)$-$C(sp^3)$ cross-coupling with secondary alkyl halides has historically suffered from predominant base promoted elimination or transition metal catalyzed β-hydride elimination pathways, we initially anticipated that the cross-coupling reaction between 8 and diphenylzinc would be low-yielding. Through careful fine-tuning of the conditions we noticed, however, that alkene formation can be overcome and we were able to achieve 73% conversion toward 10 at 70 °C (Fig. 2B). As expected, activation of the primary alkyl fluoride 9 requires higher temperatures but proceeds with almost identical conversion to 11.

With an optimized set of conditions in hand, we continued with the evaluation of the alkyl fluoride substrate scope (Fig. 2C). We discovered that unactivated primary alkyl fluorides as well as propargylic and benzylic fluorides (see products 11, 14–16) are viable candidates for the cross-coupling with diphenylzinc. This also proved to be the case when we explored several secondary alkyl fluorides (10, 17–22). The formation of 19 and 20 in 75% and 72% yield, respectively, is particularly noteworthy and compares very well with previously reported Suzuki and Kumada-Corriu reactions. In fact, the benzylic fluorides used are known to give very low yields in transition metal catalyzed cross-couplings due to predominant HF elimination toward stable styrene derivatives[31]. Furthermore, rearrangements of propargylic substrates to allenyl compounds were not observed. Next, we investigated a variety of tertiary alkyl fluorides and found that C-C bond formation occurs in good to high yields (4, 7, 23–27). Adamantyl fluoride gave 23 in 91% yield and the sterically encumbered tetraaryl products 4 and 26 where isolated in 73% and 91% yield, respectively, which shows that this reaction enables construction of highly congested scaffolds. Even when facile HF elimination would be expected to prevail due to formation of conjugated double bonds or release of steric repulsion we observed successful cross-coupling (24 and 25). We also realized that acyl fluorides, which have become popular starting materials in chemical synthesis, can be used (28–30). Altogether, these results demonstrate extraordinary resilience against hard-to-control side reactions that often render C-C bond formation with alkyl halides impractical while high functional group tolerance typical for organozinc cross-coupling chemistry is maintained. Preliminary attempts to use organosilicon, boron and tin compounds under similar conditions were less successful but demonstrate the prospect of widely useful transition metal catalyst-free organometallic $C(sp^3)$-F bond functionalization (see Supplementary Table 4).

### Mechanistic investigations

Initially, we expected that the use of weakly polar, non-coordinating solvents would be crucial to maintain the organozinc fluorophilicity but likely hamper subsequent aryl transfer due to formation of

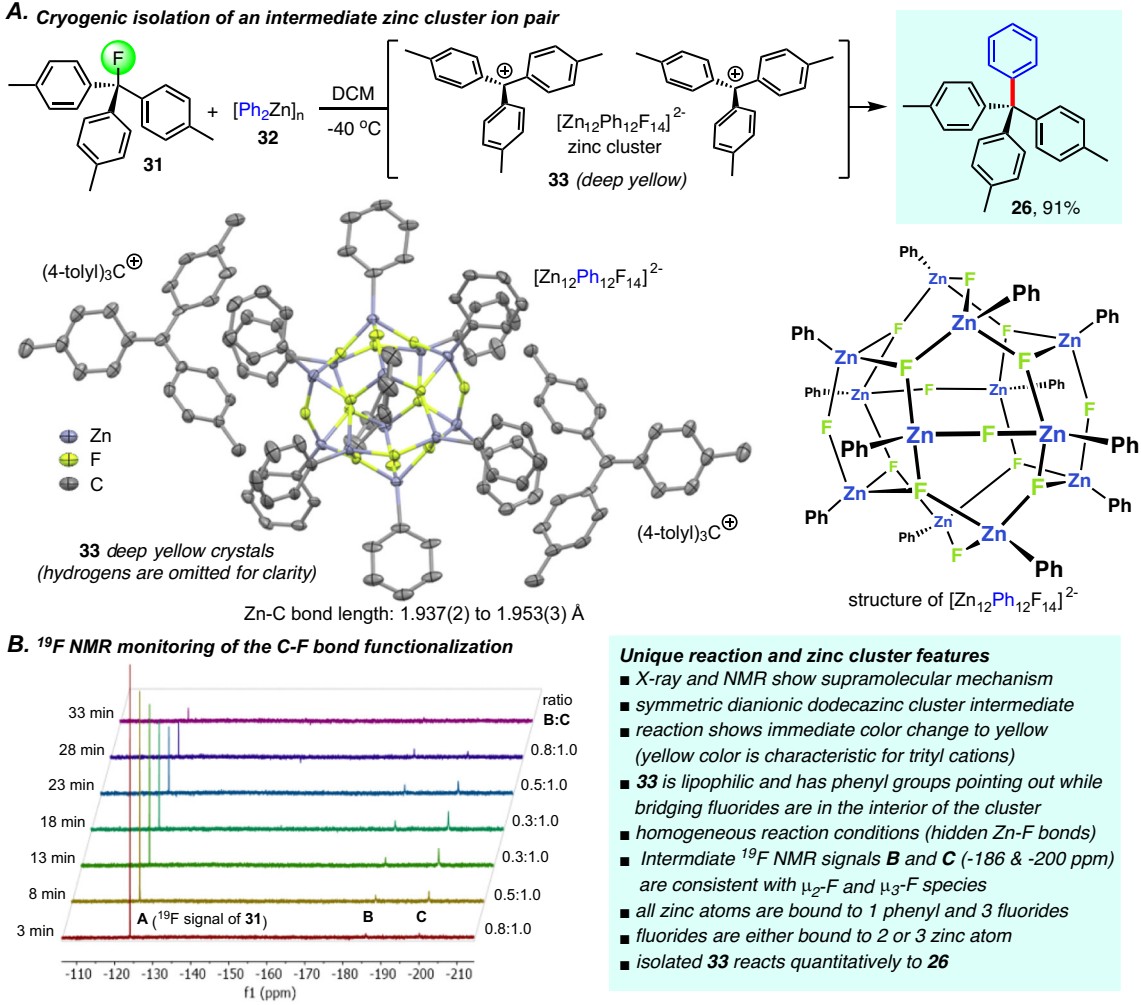

**A.** *Cryogenic isolation of an intermediate zinc cluster ion pair*

(4-tolyl)₃C⊕  [Zn₁₂Ph₁₂F₁₄]²⁻

**33** deep yellow crystals
(hydrogens are omitted for clarity)

Zn-C bond length: 1.937(2) to 1.953(3) Å

structure of [Zn₁₂Ph₁₂F₁₄]²⁻

**B.** *¹⁹F NMR monitoring of the C-F bond functionalization*

**Unique reaction and zinc cluster features**
- X-ray and NMR show supramolecular mechanism
- symmetric dianionic dodecazinc cluster intermediate
- reaction shows immediate color change to yellow
  (yellow color is characteristic for trityl cations)
- **33** is lipophilic and has phenyl groups pointing out while
  bridging fluorides are in the interior of the cluster
- homogeneous reaction conditions (hidden Zn-F bonds)
- Intermdiate ¹⁹F NMR signals **B** and **C** (-186 & -200 ppm)
  are consistent with μ₂-F and μ₃-F species
- all zinc atoms are bound to 1 phenyl and 3 fluorides
- fluorides are either bound to 2 or 3 zinc atom
- isolated **33** reacts quantitatively to **26**

**Fig. 3 | Investigation of complementary activation of aryl-zinc and C(sp³)-F bonds. A** Isolation and characterization of an intermediate zinc cluster [Zn₁₂Ph₁₂F₁₄]²⁻. Crystallographic details: Range of Zn-C bonds: 1.937(2) (shortest) to 1.953(3) (longest); The angle of μ₂-fluorides bridging two zinc atoms Zn-F-Zn (6 present in **33**) is 149.07(10)° which is very similar to the Ph₂Zn-F-ZnPh₂ species. The angles of μ₃-fluorides bridging three zinc atoms (8 present in **33**) are roughly 120°, these fluorides are virtually coplanar with their three surrounding zinc centers. **B** ¹⁹F NMR analysis of the same reaction in CDCl₃. The components were mixed at −40 °C and an immediate color change from colorless to bright yellow was observed. The resulting homogenous yellow solution was then quickly transferred to an NMR tube, sealed and monitored by ¹⁹F NMR spectroscopy every 5 min at room temperature until completion. DCM dichloromethane.

insoluble zinc fluoride salts. This was a major concern because heterolytic C–F bond cleavage followed by sluggish C(sp²)-C(sp³) bond formation would favor the competing side reactions mentioned above. In contrast, we generally observed homogeneous reaction conditions pointing toward a concerted or radical mechanism with uncharged intermediates. We noticed during method development, however, that the cross-coupling of diphenylzinc and triarylmethyl fluorides concurs with a transient color change to yellow, indicative of the presence of intermediate trityl cations, while the solution is colorless at the beginning and at completion of the reaction. Assuming that the reaction between **31** and **32** presents a unique opportunity to identify intermediate arylzinc fluoride species and to shed light on some of the unusual features of the C(sp³)-F bond functionalization, we set out to isolate single crystals under cryogenic conditions. Indeed, we were able to halt the reaction at −40 °C and deep yellow single crystals were obtained (Fig. 3A and Supplementary Figs. 10–13). Crystallographic analysis revealed heterolytic fluoride abstraction generating trityl cations and a highly symmetric dianionic dodecanuclear zinc cluster **33** thus providing a distinctive snapshot of the fluoride/aryl exchange mechanism. A closer inspection of the structure of [Zn₁₂Ph₁₂F₁₄]²⁻ shows how this is achieved. The zinc cluster shields all polar Zn-F

bonds inside the inorganic interior and exhibits a perfectly lipophilic exterior shell that makes it soluble in weakly polar organic solvents. The Zn-aryl moieties are exposed on the outside and available for the C-C bond forming step whereas the fluorides are hidden inside and not available to regenerate the C-F bond. This is an important mechanistic feature of the supramolecular pathway as the dinuclear structure **1** favors C–F bond reduction over the desired C-C bond formation in toluene and dichloromethane and does not outcompete re-fluorination of the trityl ion **3** to **31** (Fig. 1C and Section 2 in the Supplementary Information). The cluster structure is therefore conducive for the C-C bond formation to prevail over dehydrodefluorination and reversible fluorination reactions and it explains some of the unique mechanistic aspects of C-F bond functionalization via carbocation-zincate ion pair intermediates. Importantly, we found that isolated **33** continues to produce **26** which proves that the isolated zinc cluster is a plausible intermediate that is likely to co-exist with other ion-pairing species during the reaction. In fact, it is important to note that 12 arylations have already been performed before the formation of **33** is complete, i.e. the reaction is close to completion when the cluster is present at a concentration that is sufficient to allow its crystallization. It is therefore possible that a majority of the reaction proceeds through

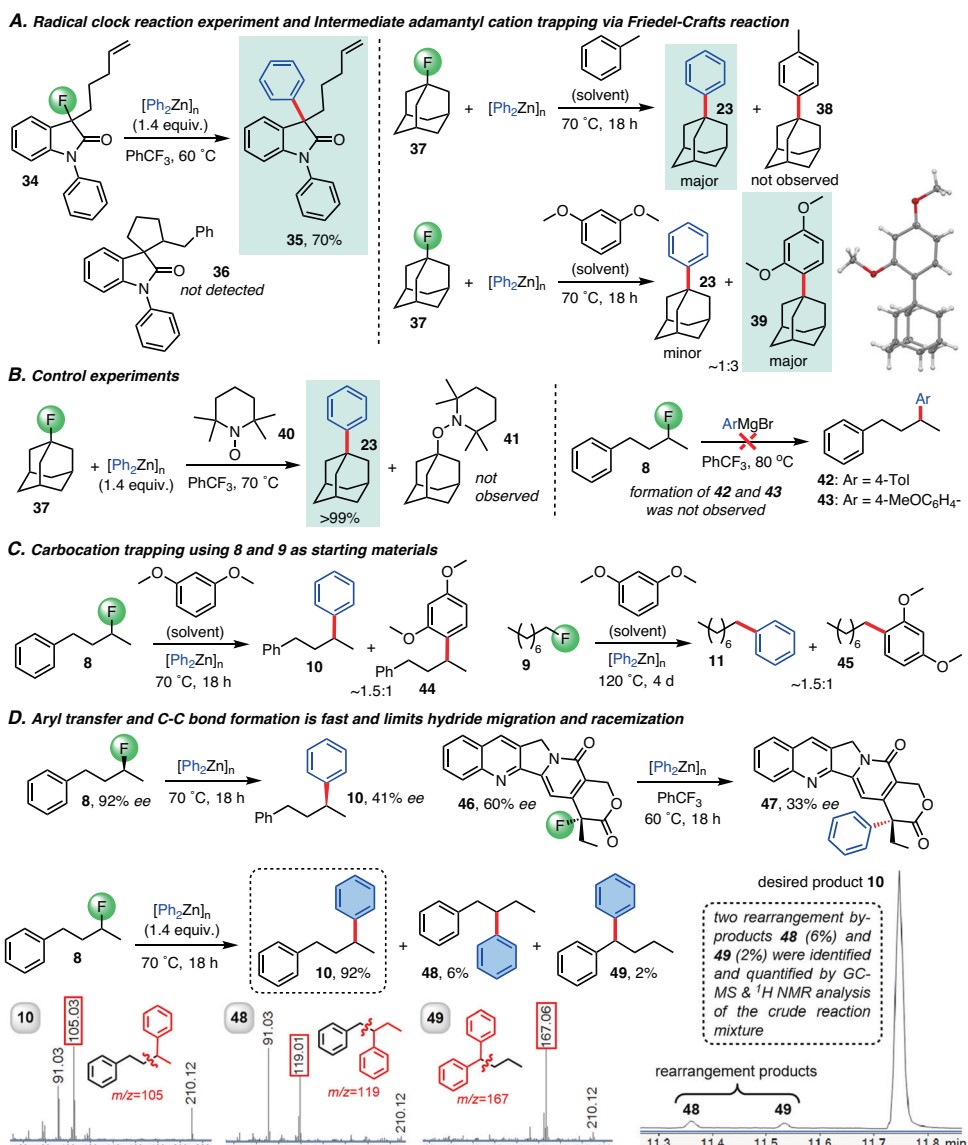

**Fig. 4 | Mechanistic studies of the alkyl fluoride cross-coupling reaction with diarylzinc compounds. A** Radical clock reaction experiment and intermediate adamantyl trapping. **B** TEMPO radical trapping does not affect the reaction outcome and the corresponding Grignard reaction did not result in C-F bond functionalization. **C** Primary and secondary alkyl fluorides also form carbocation zincate intermediates. **D**. The absolute configuration of (*S*)-**10** was assigned by comparison with the literature and that of (*R*)-**47** by analogy. The *ee* values were determined by chiral GC and HPLC analysis. Competition between desired arylation and Wagner-Meerwein rearrangements under general reaction conditions. The relative distribution among **10**, **48** and **49** is shown, the total conversion from **8** is 73%. The isomers were identified and quantified by GC-MS and ¹H NMR analysis. The most indicative MS fragmentation for each isomer is shown. equiv. equivalents.

similar aggregates that are more reactive and cannot be isolated even under cryogenic conditions. We then monitored this reaction by ¹⁹F NMR spectroscopy to determine if structures like **33** are also present in solution (Fig. 3B and Supplementary Fig. 14). A solution of **31** and **32** was prepared at −40 °C and then allowed to slowly warm to room temperature. As the signal of the trityl fluoride starting material at −124.1 ppm disappeared two transient, relatively broad upfield-shifted signals were observed. Few examples of organometallic structures with a Zn(μ₂-F) motif exhibiting a fluoride in a bridging 3c-2e bond are known but the ¹⁹F NMR shift at −200.1 ppm is in excellent agreement with literature values[43]. The unusual Zn(μ₃-F) component can be assigned to the downfield-shifted ¹⁹F NMR signal at −186.1 ppm. The X-ray and ¹⁹F NMR results thus confirm a unique supramolecular mechanism enabling heterolytic C−F bond functionalization under homogeneous conditions. We assume that the reactions with primary and secondary alkyl fluorides proceed through similar but less stable

and therefore increasingly short-lived ion-pairing intermediates that evade NMR and X-ray analysis. Additional cryogenic crystallization experiments with **31** and **32** repeatedly yielded **33** whereas other alkyl fluorides gave gels probably consisting of less-ordered aggregates.

These insights into the C-F bond functionalization reaction prompted mechanistic studies with other alkyl fluorides (Fig. 4A, B, Supplementary Fig. 15–20, 25). A clock reaction experiment with **34** showed no evidence for a radical mechanism and we isolated **35** in 70% yield. We note that Gomberg's dimer which would be expected from a radical mechanism was not observed in any of the trityl fluoride cross-couplings nor did we observe dimerization of any of the substrates used in this study. We then attempted to trap the intermediate adamantyl cation generated by C-F abstraction of **37** via Friedel-Crafts alkylation using toluene as solvent. Surprisingly, this gave **23** as the major product and no sign of **38** was observed by GC-MS. The fact that the C-F bond functionalization quantitatively outcompetes the

*A. Features of C-F bond functionalization with substituted (hetero)arylzinc compounds*

- dual role of Ar$_2$Zn: fluorophilic Lewis acid & aryl transfer agent
- wide substrate scope: 1°, 2°, 3° alkyl fluorides & sterically hindered (hetero)aryl Zn reagents
- selective C(sp$^3$)-F bond activation, wide functional group tolerance, access to crowded tetrasubstituted carbon scaffolds
- homogeneous conditions, solvent choices include PhCF$_3$, CH$_2$Cl$_2$, CHCl$_3$, or solvent-free
- short-lived ion pair intermediate is crucial for C-C bond formation, outcompetes rearrangements, HF elimination, hydrodefluorination

*B. Late-stage functionalization*

**Fig. 5 | Scope of alkyl fluoride cross-coupling with various diarylzinc compounds. A** C−F bond functionalization with sterically hindered aryl- and heteroarylzinc compounds. **B** Demonstration of late-stage functionalization. All yields are based on isolated products. equiv. equivalents.

trapping reaction in toluene is remarkable because it demonstrates that the C-C bond formation step is fast and occurs almost instantaneously inside the solvent cage. Only when we resorted to the more reactive 1,3-dimethoxybenzene as solvent we obtained considerable amounts of the Friedel-Crafts product **39**. As expected, addition of the radical scavenger TEMPO did not affect the reaction outcome and the corresponding Grignard reactions with **8** were unsuccessful. We then turned our attention to primary and secondary alkyl fluorides. Friedel-Crafts trapping experiments using **8** and **9** in 1,3-dimethoxybenzene gave **44** and **45**, respectively, as expected for an ionic mechanism with intermediate carbocation-zincate ion pair generation (Fig. 4C and

Supplementary Figs. 21–24). We obtained the enantioenriched secondary alkylfluorides **8** and **46** to provide further mechanistic insights into the C-C bond formation process. We observed considerable erosion of the *ee*'s in the reactions with diphenylzinc but not full racemization (Fig. 4D, Supplementary Fig. 26–36). Comparison with literature reports revealed that the reaction of (*S*)-**8** (92% *ee*) proceeds with retention of configuration and we isolated (*S*)-**10** in 41% *ee*. The reaction with **46** (60% *ee*) gave **47** in 33% *ee*. These results suggest that the aryl nucleophile delivery in the intermediate ion pair is fairly fast and able to compete with molecular tumbling processes that are likely to cause the erosion in the compound enantiopurities.

The effect of solvent polarity on the reaction between 1-fluorooctane, **9**, and Ph$_2$Zn was monitored by GC-MS analysis (see Section 2 in the Supplementary Information). Because the fluorophilicity of organozinc compounds is sensitive to Lewis-basic solvents we were limited in our selection and chose hexafluorobenzene ($\varepsilon = 2.05$) and trifluorotoluene ($\varepsilon = 9.40$) for this study. We observed no discernable reactivity trend with respect to solvent polarity and both reactions showed almost identical conversion at various time intervals. These results are in agreement with the assumption of lipophilic, short-lived carbocation-zincate ion pairs that are soluble in weakly polar solvents. Because the uncharged ion pairs react very quickly within the solvent shell to the desired cross-coupling products solvent stabilization effects are negligible. The analyses shown in Figs. 3 and 4 thus consistently corroborate that this C(sp$^3$)-F bond functionalization method is mechanistically different from previously reported radical reactions[33–35,45]. The general success with unactivated alkyl fluorides prone to rearrangements, HF elimination and hydrodefluorination and the failure of the Friedel-Crafts trapping in toluene encouraged us to quantify isomeric by-product formation using the secondary alkyl fluoride **8** (Fig. 4D). We obtained 90% of the desired coupling product **10** and the rearrangement products **48** and **49** in only 6% and 2%, respectively. The predominance of C-F bond functionalization over intrinsically fast rearrangements and HF elimination is consistent with the assumption of a spontaneous aryl transfer that outpaces intramolecular processes. This was also observed with other challenging primary, secondary and tertiary substrates or benzylic fluorides that are known to rapidly form styrene derivatives (for example, see products **7, 11, 19, 20, 24** and **25** in Fig. 2). Wagner-Meerwein rearrangement by-products were completely absent when electron-rich organozinc reagents expected to further reduce the ion pair lifetime were employed (*vide infra*).

## Alkyl fluoride cross-coupling with substituted (Hetero)arylzinc compounds and late-stage functionalization

We then decided to further expand the synthetic scope and explored organozinc compounds carrying electron-deficient or electron-rich (hetero)arenes as well as very bulky mesityl rings, which all proved to be viable for C(sp$^3$)-F bond functionalization (Fig. 5A). When we investigated the cross-coupling of unactivated primary alkyl fluorides with bis(benzothiophene)zinc or other electron-rich reagents we did not observe any sign of rearrangement by-product formation (for example, see **51** and **52**). Apparently, the transfer of electron-rich aryl rings is increasingly fast and quantitatively outcompetes hydride shifts. We then achieved C-F bond activation and transfer of a ferrocenyl moiety, producing **53** in 71% yield. The practicality and ease of these transformations then prompted us to employ other organozinc compounds carrying heteroaryl or highly substituted rings that either pose electronic or steric challenges. Using several secondary alkyl fluorides that are poor starting materials in currently available Kumada-Corriu or Suzuki reaction procedures we observed successful (hetero)arylation toward a variety of C(sp$^3$)-C(sp$^2$) coupling products (**54–58**). Similar transformations with tertiary substrates were equally straightforward and conquer additional synthetic space for organometallic alkyl fluoride transformation (**59–69**). The construction of highly congested scaffolds like **59** which was obtained from adamantyl fluoride and bis(mesityl)zinc in 75% yield is particularly noteworthy and demonstrates that this methodology tolerates considerable steric bulk in both the alkyl fluoride and the zinc reagent and overcomes severe problems with overwhelming side reactions often encountered during transition metal catalyzed C-C bond formation.

Perhaps the most impressive illustration of the general utility and synthetic prospects of this chemistry is embodied by late-stage functionalization examples that capture the striking functional group compatibility, chemoselectivity and stereocontrol accessible through C(sp$^3$)-F bond functionalization of pharmaceutically or agriculturally relevant compounds (Fig. 5B). We discovered that phenylation of fluoroquinine **70** gives **71** in 65% yield and occurs with complete retention of configuration. The same stereoselectivity was observed in the reaction of **70** with di(4-chlorophenyl)zinc toward the alkaloid **74**. The stereochemical outcome of these reactions was verified by NMR spectroscopy and X-ray analysis using single crystals of the corresponding hydrochloride salts. The realization of highly stereocontrolled C(sp$^3$)-F bond functionalization further corroborates our mechanistic analysis showing that the intermediate ion pairs are short-lived and react rapidly in the solvent cage compared to diffusion and molecular tumbling processes that would compromise the stereochemical purity of the products. While **70** is expected to exhibit intrinsic diastereoselectivity, its relatively large size effectively impedes tumbling motions that could change the relative orientation between the carbocation and the zincate prior to the C-C bond formation and therefore favors fluoride replacement with retention of configuration. Using our previously developed protocols without further optimization we were also able to transform fluoronuarimol and fluorodonepezil to **72** and **73** in 82% and 75% yield, respectively. Finally, the known drug fluorocamptothecin[46] was converted to the tetra-substituted lactone **47**.

In conclusion, we have introduced a C(sp$^3$)-F bond functionalization method that overcomes long-standing limitations of organometallic cross-coupling chemistry, including predominant HF elimination, homodimerization or hydrodefluorination side reactions. This is achieved with a thoroughly investigated reciprocal activation concept by which C-F bond cleavage generates lipophilic carbocation-zincate ion pairs that are soluble in apolar solvents and exhibit fast aryl transfer kinetics successfully outcompeting rearrangement reactions while the stability of the Zn-F bond serves as the ultimate thermodynamic sink. This approach is broadly useful and significantly expands the currently limited synthetic space of unactivated primary, secondary and tertiary C(sp$^3$)-F bonds while benzylic, propargylic or acyl fluorides can also be employed. The application scope includes challenging substrates with high propensity for HF elimination and hydrodefluorination reactions which widely precludes their usage in conventional alkyl halide cross-couplings. In addition, a large variety of functionalities (for example, nitrile, ketone, amide, ester, pyridine, furan, thiophene, lactam, lactone, nitro, carbonate, CF$_2$, CF$_3$, aryl halides including C(sp$^2$)-F, alkene, alkyne, ether, amino groups) and sterically demanding substrates are tolerated, enabling late-stage functionalization of pharmaceutically and agrochemically relevant scaffolds. As with any method, there are remaining limitations and attempts to use substrates containing free alcohol or acyclic ester groups were not successful. Selective C(sp$^3$)-F bond functionalization with organozinc compounds is not only synthetically attractive by achieving formal Negishi coupling without the need for expensive catalysts that may necessitate not readily available, specialized ligands and are often of limited use due to notorious complications from competing β-hydride elimination and homocoupling pathways. It also introduces opportunities for synthetic chemists, for example, a retrosynthetic disconnection approach that exploits the C-F bond as a placeholder inert under traditional synthesis conditions but available for late-stage derivatizations. The discovery of a supramolecular mechanism involving short-lived ion-pairing zincates with a lipophilic exterior adapted to low-dielectric solutions has wider implications and is expected to stimulate the development of methods that utilize the distinct propensity of fluoride to generate synthetically useful higher-order reaction intermediates.

## Methods
### General information
Commercially available organofluorines (1-fluorooctane, 1-fluorododecane, 4-fluoro-1,3-dioxolan-2-one, 1-fluoroadamantane,

3-chloro-2-fluorobenzoyl fluoride, 4-(trifluoromethyl)benzoyl fluoride, benzoyl fluoride, and 4-bromo-1-(fluoromethyl)−2-methylbenzene) and organozinc compounds were used as purchased without further purification. Solvents were stored over 4 Å molecular sieves prior to use. All reaction products were purified by column chromatography on silica gel (particle size 40–63 µm) as described below. Air-sensitive reactions were carried out in a nitrogen-filled glovebox or with standard Schlenk techniques. NMR spectra were obtained at 400 MHz ($^1$H NMR), 100 MHz ($^{13}$C NMR) and 376 MHz ($^{19}$F NMR) and chemical shifts were referenced to residual solvent peaks. GC-MS measurements were acquired on an Agilent 5977 C GC/MSD equipped with an HP-5ms Ultra Inert (5%-phenyl)-methylpolysiloxane column (30 m, 0.25 mm, 0.25 µm). Chiral GC analysis was performed on a 2,6-dimethyl-3-pentyl-γ-cyclodextrin chiral stationary phase. Chiral HPLC was conducted using CHIRALCEL OD-H as chiral stationary phase. HRMS data were obtained using electron spray ionization time-of-flight (ESI-TOF) spectrometry.

### X-ray crystallography

Single crystals of each compound were mounted under parabar oil on a Mitegen micromount and immediately placed in a cold nitrogen stream at 100(2) K prior to data collection. Data were collected on either a Bruker D8 Quest equipped with a Photon100 CMOS detector and a Mo ImS source or a Bruker DUO equipped with an APEXII CCD detector and Mo fine-focus sealed source. Data were integrated with the Bruker SAINT program. Structure solution and refinement was performed using the SHELXTL/PC suite and ShelXle. Intensities were corrected for Lorentz and polarization effects and an empirical absorption correction was applied using Blessing's method as incorporated into the program SADABS. Non-hydrogen atoms were refined with anisotropic thermal parameters. Hydrogen atoms were included in idealized positions unless otherwise noted (Supplementary Figs. 183–194).

### Synthesis

Triphenylfluoromethane, racemic and enantiomerically enriched (S)-(3-fluorobutyl)benzene (92% ee), 1-fluorodec-2-yne, 1-(fluoromethyl)−4-nitrobenzene, 1-(3-fluorobutyl)−4-methoxybenzene, 1-(1-fluoroethyl)−3-nitrobenzene, (3-fluorobut-1-yn-1-yl)benzene, (cyclohexylfluoromethyl)benzene, 4-(fluoro(2-tolyl)methyl)benzonitrile, 2-(1-fluoro-1-phenylethyl)pyridine, tri(4-tolyl)fluoromethane, (1 S,2 S,4 S,5 R)−2-((S)-fluoro(6-methoxyquinolin-4-yl)methyl)−5-vinylquinuclidene, 1-bromo-4-(fluoro(phenyl)methyl)benzene, 5-((2-chlorophenyl)fluoro(4-fluorophenyl)methyl)pyrimidine, 4-ethyl-4-fluoro-1,12-dihydro-14H-pyrano[3′,4′:6,7]indolizino[1,2-b]quinolin-3,14(4H)-dione (60% ee), 2-(3-fluoropropyl)isoindoline-1,3-dione, and 5-(3-fluorobutyl)benzo[d][1,3]dioxole were prepared in one step from the corresponding alcohol using diethylamino sulfurtrifluoride according to the literature[47]. 3-Fluoro-3-methyl-1-phenylindolin-2-one, 2-fluoro-2-methyl-1-phenylpropan-1-one, and 2-((1-benzylpiperidin-4-yl)methyl)−2-fluoro-5,6-dimethoxy-2,3-dihydro-1H-inden-1-one were prepared in one step using electrophilic fluorination according to the literature[48]. Experimental details of alkyl fluoride and organozinc compound preparations, C(sp$^3$)-F bond functionalization methods, product purification and characterization are provided in Sections 5.1. to 5.4. of the Supplementary Information. Copies of NMR spectra of all compounds can be found in Section 6 of the Supplementary Information (Supplementary Figs. 38–182).

### Data availability

The authors declare that the data supporting the findings of this study are available within the paper and its Supplementary Information. All data are available from the corresponding author upon request. CCDC 2220190 (Fc$_2$Zn), CCDC 2261444 ([Ph$_2$Zn]$_3$), CCDC 2261445 (1), CCDC 2261447 (2), CCDC 2220186 (7), CCDC 2220184 (33), CCDC 2220185 (39), CCDC 2220188 (62), CCDC 2220191 (63), CCDC 2220422 (66), CCDC 2220189 (71), and CCDC 2220187 (74) contain the supplementary crystallographic data for this paper. These data can be obtained free of charge from The Cambridge Crystallographic Data Centre.

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

## Acknowledgements

We gratefully acknowledge financial support from the US National Institutes of Health, GM106260 (CW).

## Author contributions

C.W., D.L.K., B.C.F., K.B. and J.A.B. designed the experiments and analyzed the data. D.L.K., B.C.F. and K.B. performed all experiments. C.W. and D.L.K. wrote the manuscript and supplementary information. C.W. conceived and supervised the project. All authors discussed the results and commented on the manuscript.

## Competing interests

The Authors declare no competing interests.
