## [Peer Review File · Nature Communications]

REVIEWER COMMENTS

Reviewer #1 (Remarks to the Author):

This paper describes the arylation of inactive C(sp³)-F bonds by cleavage with diarylzinc reagent. It reveals a new chemical species, the previously reported diarylzinc reagent being a trimeric aggregate. Furthermore, the formation of the Zn-F bond is the driving force behind this arylation, and the finding that the diarylzinc reagent causes the fluorine atom to be trapped via heterolytic cleavage to form an intermediate zinc cluster ion pair is also significant. This transformation, which can tolerate various functional groups, is a breakthrough, as it is distinct from the reaction mechanism using transition metal catalysts.

However, on the other hand, the reaction itself is specific to C(sp²)-C(sp³) bond formation and can be regarded as routine work in that similar reaction products can be synthesized in other Kumada-Corriu and Suzuki type cross-coupling reactions and transition metal-catalyzed Negishi couplings.

The reviewer is therefore reluctant to accept this paper for publication in Nature Communications.

The authors are recommended to resubmit their work to other journals with the following comments.

- 1) Trifluorotoluene is not a practical and inexpensive organic solvent.
- 2) The two structures and ORTEPs of compound 7 are shown in Figure 2. In the Table, the yield of compound 7 is 80%, while in C it is 68%. Which value is correct?
- 3) Why does the ratio of the B and C signals change over time in the NMR in Figure 3B? From the structure of compound 33, the ratio should be B:C = 6:8 (=3:4).
- 4) The hydride migration shown in Figure 4C is an interesting consideration from a scientific point of view, but a disadvantage from a synthetic point of view. It is also not consistent with the results in the Table in Figure 2B and the content should be checked again.
- 5) "Tempo" should be "TEMPO" and "benzothiophen" should be "benzothiophene".
- 6) The configuration is retained when compound 67 is formed from compound 66, why is this? The reason should be added in the text.

Reviewer #2 (Remarks to the Author):

The authors of "General Alkyl Fluoride Functionalization with Short-lived Ion-pairing Organozincate Clusters" describe a diaryl zinc arylations of CF bonds in fluoroorganics. We try to remove fluoroorganics from our chemical applications as much as we can (especially the polyfluorinated ones) but the formidable challenge that their activation can pose keeps this study goal very interesting. The reaction can address a rather broad scope, both in regard to the activated CF bond and the transferred aryl. The authors find a polar pathway over the often seen radical one. The mechanistic distinction to typical Ar₂Zn reagents is exciting but there is no direct connection between the cluster formation and the good reaction selectivity. The authors provide thorough insights into the discussed chemistry with interesting

and technically well-performed control experiments. The observed differences to established chemistry could (potentially) be justified by the CF bond being different to previously activated bonds in bond order, polarity, and redox potential. For this reason, I suggest that some of the claims should be worded more carefully or justified experimentally/computationally. I recommend reconsideration after major corrections.

The scope is broad but a brief comment should be made on limitations of the reaction. Alcohols? Acyclic esters?

The introduction sets the stage in perhaps a more broad context than needed. Only when the discussion starts, references to established zinc(ate) chemistry are made. This is done after the introduction ended with a summary of this work. The introduction does not deliver a reason for why zinc would be used and a state of the art regarding the choice. Some rearrangements would help here to get the readers to not miss the points regarding the rarity of ionic pathways and fluorozincates.

The figures seem crowded. Some whitespace, particularly in fig 1, would make digesting the figures less demanding. Parts of the bullet points could be shortened or left out where they are just pointing out things that are very obviously displayed in the figure anyways. For example, do we need to write wide scope, functional group tolerance, and so on in fig. 2 when the scope is shown.

The reactions of 1/2 with 3 to show selectivity differences regarding arylation and reduction were done in different solvents. Presumably, the hydride found in the reduction product is sourced from the solvent. Data in the same solvent should be provided. These experiments comparing a dinuclear Ar_2Zn and a oligonuclear $[\text{Ar}_3\text{ZnLi}]$ complex are later used directly to justify the observed reactivity of the $[\text{Zn}_{12}\text{Ph}_{12}\text{F}_{14}]^{2-}$ cluster. With charge and coordination environment being distinct, such comparisons should be made more carefully.

Fig. 3 gives the statement that 33 reacts quantitatively to 26. The figure contains this reaction. It has a 91% yield given. The text seems inaccurate and unnecessary.

The authors should describe where the additional 12 fluorides in compound 33 are sourced. Only two out of fourteen trityls in the production of the crystalline intermediate are accounted for. The implications of this should be explained experimentally or verbally. Apparently 12 arylations have already been performed before the formation of 33 is complete, which makes 33 as intermediate questionable. Does this not suggest that a majority of the reaction would not go via this cluster and that just the arylation starting from this cluster is the slowest which makes it isolable?

There is limited data present to support the same mechanism being active in primary fluorides. A concerted mechanism may become active where fluoride abstraction becomes challenging. This should be elaborated experimentally, computationally, or verbally. A possible experiment avoiding direct cation detection could be kinetic measurements in solvents of different polarity. If the ionic intermediate would be part of the rate determining path, polar solvents should significantly increase the reaction rate. A concerted mechanism should be less affected.

Which intermediate zinc cluster is shown in fig. 5A? The bullet points do not seem to explain why the cluster is shown here. The bullet point claims that the cluster ion pair is crucial for a number of reasons which are likely but not proven. The CF bonds themselves are different to previously activated bonds. It seems if the cluster formation is the reason for the distinct behaviour. Showing and discussing the cluster

again in this figure is only repetition of previously displayed and discussed items and could be avoided to avoid confusion.

Reviewer #3 (Remarks to the Author):

The manuscript by Wolf describes a catalyst-free cross-coupling between alkyl fluorides and diaryl zinc species. Various alkyl fluorides are applied to the reaction, including primary, secondary, and tertiary alkyl fluorides. In particular, unactivated alkyl fluorides are good coupling partners for the current process, in sharp contrast to the previous reports that were limited to narrow substrate scope. Mechanistic studies reveal that short-lived ion-pairing organozincate clusters are responsible for the reaction, and a unique supramolecular reaction pathway enables rapid aryl transfer. This is a nice contribution to the C(sp²)-C(sp³) bond formation, which overcomes the limitations of transition-metal-catalyzed coupling, including β -hydride elimination and homodimerization. However, the substrate scope and functional group tolerance are not as good as the authors emphasized. Most substrates examined in the reaction are activated, such as propargyl and benzyl fluorides. The reaction tolerates aryl bromide, cyanide, and acetamide, but did not exhibit high functional group tolerance. Most importantly, all the alkyl fluorides are not readily available as the authors mentioned, which would regulate the synthetic utility of the approach. Overall, given the detailed mechanistic studies and the novel pathway involved in the reaction, the manuscript could be published in Nat. Commun. after revision.

1) The use of lithium salt to activate the C-F bond for functionalization is well-known. In one example Figure 2B, the addition of LiI to the reaction benefits the reaction efficiency (10); however, for substrate 9, poor yield was obtained by using LiI. Comments on these differences should be provided.

2) The preparation of alkyl fluorides should be clarified in the main text and SI. If the authors used a tedious procedure to prepare alkyl fluorides, it would decrease the synthetic utility of this approach.

3) Typos

"(Fig. 2C and SI)" should be "(Fig. 1C and SI)."

Dear Reviewers:

We appreciate the very helpful comments and suggestions and have substantially revised our manuscript as described in detail below in our point-by-point responses. We have made major changes, in particular with regard to the mechanism where we now include additional work with primary and secondary alkyl fluorides, but also in the introduction to improve the flow of the manuscript. A copy of the manuscript with changes highlighted in yellow is included (editorial changes in the manuscript, for example, compound renumbering, reference reformatting and changes/additions in the SI are not highlighted for clarity).

Reviewer #1 (Remarks to the Author):

This paper describes the arylation of inactive C(sp³)-F bonds by cleavage with diarylzinc reagent. It reveals a new chemical species, the previously reported diarylzinc reagent being a trimeric aggregate. Furthermore, the formation of the Zn-F bond is the driving force behind this arylation, and the finding that the diarylzinc reagent causes the fluorine atom to be trapped via heterolytic cleavage to form an intermediate zinc cluster ion pair is also significant. This transformation, which can tolerate various functional groups, is a breakthrough, as it is distinct from the reaction mechanism using transition metal catalysts.

However, on the other hand, the reaction itself is specific to C(sp²)-C(sp³) bond formation can be regarded as routine work in that similar reaction products can be synthesized in other Kumada-Corriu and Suzuki type cross-coupling reactions and transition metal-catalyzed Negishi couplings.

The reviewer is therefore reluctant to accept this paper for publication in Nature Communications.

We appreciate that this Reviewer commends the alkyl fluoride functionalization with organozinc compounds described in our paper as a “breakthrough”. We like to add that our findings have various synthetic and mechanistic implications. Despite the wide occurrence of the increasingly ubiquitous C(sp³)-F bond, synthetically useful functionalizations are scarce and current possibilities are trailing far behind the striking synthetic utility of alkyl chlorides, bromides and iodides. We introduce practical C_{sp3}-F bond functionalization with equimolar amounts of organozinc reagents, addressing long-standing problems of organometallic C(sp²)-C(sp³) cross-coupling with alkyl fluorides, including predominant side reactions such as HF elimination, homodimerization or hydrodefluorination. This is achieved with a thoroughly investigated reciprocal activation concept (we now include additional mechanistic studies) by which C-F bond cleavage generates lipophilic carbocation-zincate ion pairs exhibiting fast aryl transfer kinetics that outcompete rearrangement reactions while the stability of the Zn-F bond serves as the ultimate thermodynamic sink. This chemistry is mechanistically different from couplings of arylzinc compounds with other alkyl halides, broadly useful and significantly expands the currently limited synthetic space of unactivated primary, secondary and tertiary C(sp³)-F bonds while benzylic, propargylic or acyl fluorides can also be employed. In addition,

a large variety of functionalities (for example, nitrile, ketone, amide, ester, pyridine, furan, thiophene, lactam, lactone, nitro, carbonate, CF₂, CF₃, aryl halides including C(sp²)-F, alkene, alkyne, ether, amino groups) and sterically demanding substrates are tolerated, enabling late-stage functionalization of pharmaceutically and agrochemically relevant scaffolds. In fact, our work largely outperforms the scope and functional group tolerance of Kumada-Corriu cross-coupling. Alkyl fluoride Suzuki couplings are also of limited scope and couplings with Zn systems have not been reported at all.

We agree with this Reviewer that similar products can also be prepared by Suzuki and Negishi-type cross-coupling reactions with alkyl chlorides, bromides or iodides. However, these functionalities are significantly more reactive and labile than the C-F bond which is widely considered chemically inert. Our work thus provides new opportunities for synthetic chemists. For example, one can consider a retrosynthetic disconnection approach that exploits the alkyl fluoride moiety as a placeholder or protecting group that is inert (unlike other alkyl halides) under traditional synthesis conditions. One can thus subject a molecule containing a C-F bond to a variety of routine reactions until the C-F moiety is ultimately made available by our method for late-stage functionalization.

The selective C(sp³)-F bond functionalization with organozinc compounds described herein is not only synthetically attractive by achieving formal Negishi coupling without the need for expensive catalysts. The mechanism involving short-lived ion-pairs adapted to low-dielectric solutions has wider implications and is envisioned to inspire the development of new synthetic methods that utilize the distinct propensity of fluoride to generate activated higher-order organometallic intermediates as demonstrated herein. We believe that the demonstration of generally applicable C-F bond functionalization/cross-coupling with diarylzinc compounds (using unactivated primary, secondary, tertiary C-F bonds as well as activated substrates) with very broad scope, high functional group tolerance and late-stage functionalizations is likely to receive broad attention and stimulate other groups to investigate this chemistry with other organometallic compounds. To this end, we also indicate in this study that transition metal catalyst-free C(sp³)-F bond functionalization with other organometallic reagents is possible. In fact, we show in the SI that this should be possible with triarylboron compounds. To clearly highlight this in the manuscript, we state on page 7 that “Preliminary attempts to use organosilicon, boron and tin compounds under similar conditions were less successful but demonstrate the prospect of widely useful transition metal catalyst-free organometallic C(sp³)-F bond functionalization (see SI).”

The authors are recommended to resubmit their work to other journals with the following comments.

1) Trifluorotoluene is not a practical and inexpensive organic solvent.

In their analysis of trifluorotoluene as solvent Curran et al reported more than 25 years ago that it has “a relatively low toxicity and price” and “favorable properties” (J. Org. Chem. 62, 450-451 (1997)). Since then, it has become an alternative solvent to dichloromethane. A comparison at Sigma-Aldrich (on 12-20-2023) showed that trifluorotoluene is cheaper and safer than THF which is a very common solvent. The price for 1L of anhydrous trifluorotoluene was \$112 compared to \$152 for 1L of anhydrous THF. The NFPA health, flammability and instability hazard levels of trifluorotoluene are 0-3-0 whereas those of THF are 2-3-1.

2) The two structures and ORTEPs of compound 7 are shown in Figure 2. In the Table, the yield of compound 7 is 80%, while in C it is 68%. Which value is correct?

We removed the second X-ray structure. The Table reports conversion measured by GC-MS while in Figure 2C we report isolated yields.

3) Why does the ratio of the B and C signals change over time in the NMR in Figure 3B? From the structure of compound 33, the ratio should be B:C = 6:8 (=3:4).

The isolated ion pair 33 cannot be the only intermediate as many alkyl fluorides and aryl groups have already been cleaved and transferred, respectively, at this stage of the reaction. As we now state on page 8 “Importantly, we found that isolated 33 continues to produce 26 which proves that the isolated zinc cluster is a reactive intermediate that is likely to co-exist with other ion-pairing species during the reaction.” The change in the ¹⁹F NMR signal ratios shown in Figure 3B is expected as several intermediate species evolve throughout the reaction. The intermediate 33 is neither expected to persist from beginning to end nor is it the only intermediate species. Thus, the ¹⁹F NMR signal ratio should neither be constant nor necessarily reach 3:4.

4) The hydride migration shown in Figure 4C is an interesting consideration from a scientific point of view, but a disadvantage from a synthetic point of view. It is also not consistent with the results in the Table in Figure 2B and the content should be checked again.

We agree that the hydride migration is mechanistically interesting and in agreement with the proposed heterolytic C-F bond cleavage. As with any side reaction, it is synthetically a disadvantage. The study shown in Figure 4C reports the relative distribution among the three isomeric products while in the table in Figure 2B conversion to the desired product is discussed. Thanks for pointing out that this wasn't clear. We clarify this now in the caption of Figure 4D: “The relative distribution among 10, 48 and 49 is shown, the total conversion from 8 is 73%.”

5) “Tempo” should be “TEMPO” and “benzothiophen” should be “benzothiophene”.

Thanks, this has been corrected in the manuscript, it was correct in the SI.

6) The configuration is retained when compound 67 is formed from compound 66, why is this? The reason should be added in the text.

Please note that compound numbers have been changed. We greatly appreciate this very important observation and note that this is the same with product 74. The reactions with 70 (formerly 66) proceed with excellent diastereoselectivity and the retention of configuration is an important mechanistic feature that we have investigated more with a highly enantioenriched secondary alkyl fluoride (see our responses to Reviewer 2). We now explain this better on page 14: “The realization of highly stereocontrolled C(sp³)-F bond functionalization further corroborates our mechanistic analysis showing that the intermediate ion pairs are short-lived and react rapidly in the solvent cage compared to diffusion and molecular tumbling processes that would compromise the stereochemical purity of the products. While 70 is expected to exhibit intrinsic diastereoselectivity, its relatively large size effectively impedes tumbling motions that could change the relative orientation between the carbocation and the zincate prior to the C-C bond formation and therefore favors fluoride replacement with retention of configuration.”

Reviewer #2 (Remarks to the Author):

The authors of "General Alkyl Fluoride Functionalization with Short-lived Ion-pairing Organozincate Clusters" describe a diaryl zinc arylations of CF bonds in fluoroorganics. We try to remove fluoroorganics from our chemical applications as much as we can (especially the polyfluorinated ones) but the formidable challenge that their activation can pose keeps this study goal very interesting. The reaction can address a rather broad scope, both in regard to the activated CF bond and the transferred aryl. The authors find a polar pathway over the often seen radical one. The mechanistic distinction to typical Ar_2Zn reagents is exciting but there is no direct connection between the cluster formation and the good reaction selectivity. The authors provide thorough insights into the discussed chemistry with interesting and technically well-performed control experiments. The observed differences to established chemistry could (potentially) be justified by the CF bond being different to previously activated bonds in bond order, polarity, and redox potential. For this reason, I suggest that some of the claims should be worded more carefully or justified experimentally/computationally. I recommend reconsideration after major corrections.

We appreciate that this Reviewer recognizes the formidable challenge of C-F bond functionalization, the rather broad scope of our method, and the mechanistic distinction. The comment that there is “no direct connection between the cluster formation and the good reaction selectivity” was particularly helpful and prompted more mechanistic studies in our laboratory. As suggested, some of the claims including the manuscript title are now worded more carefully, removed or justified experimentally as detailed below.

The scope is broad but a brief comment should be made on limitations of the reaction. Alcohols? Acyclic esters?

That is a good point. We now added the following statement on page 15: “As with any new method, there are remaining limitations and attempts to use substrates containing free alcohol or acyclic ester groups were not successful.”

The introduction sets the stage in perhaps a more broad context than needed. Only when the discussion starts, references to established zinc(ate) chemistry are made. This is done after the introduction ended with a summary of this work. The introduction does not deliver a reason for why zinc would be used and a state of the art regarding the choice. Some rearrangements would help here to get the readers to not miss the points regarding the rarity of ionic pathways and fluorozincates.

We followed this advice and changed or rearranged the text in first few pages of the manuscript. We now mention radical $\text{C}(\text{sp}^2)$ - $\text{C}(\text{sp}^3)$ cross-couplings using organozinc and alkyl chlorides, bromides or iodides in the introduction and better prepare the reader for the rationale of pursuing a mechanistically distinct pathway, the corresponding challenges, the state of the art with Zn chemistry, the lack of knowledge on heterolytic C-F bond cleavage with diorganozinc compounds, and the rarity of any information on arylfluorozincates. We agree that this greatly improves the flow of the paper.

The figures seem crowded. Some whitespace, particularly in fig 1, would make digesting the figures less demanding. Parts of the bullet points could be shortened or left out where they are just pointing out things that are very obviously displayed in the figure anyways. For example,

do we need to write wide scope, functional group tolerance, and so on in fig. 2 when the scope is shown.

We removed or shortened some of the bullet points in the figures and increased white space instead.

The reactions of 1/2 with 3 to show selectivity differences regarding arylation and reduction were done in different solvents. Presumably, the hydride found in the reduction product is sourced from the solvent. Data in the same solvent should be provided. These experiments comparing a dinuclear Ar_2Zn and a oligonuclear $[\text{Ar}_3\text{ZnLi}]$ complex are later used directly to justify the observed reactivity of the $[\text{Zn}_{12}\text{Ph}_{12}\text{F}_{14}]^{2-}$ cluster. With charge and coordination environment being distinct, such comparisons should be made more carefully.

This was a good idea. We reran the reaction between 1 and 3 in toluene for 2 hours to have consistent conditions for the comparison of the reactions between 1 and 2 with 3. The reaction showed other by-products but the ratio of prevailing hydrodefluorination to desired arylation was similar to what we had seen in dichloromethane. Figure 1 and the text have been changed accordingly and the experiments are described in the SI. The results obtained with dichloromethane as solvent are still important because this is the same solvent used with the cluster. We believe the comparisons are now expressed more carefully in the text on pages 5 and 8.

Fig. 3 gives the statement that 33 reacts quantitatively to 26. The figure contains this reaction. It has a 91% yield given. The text seems inaccurate and unnecessary.

The statement in Fig. 3 refers to isolated 33. The 91% yield refers to the synthesis of 26 from the reaction of 31 and 32, which is different. We believe it is important that the isolated cluster 33 continues to react quantitatively to the same product showing it is a plausible intermediate in this reaction.

The authors should describe where the additional 12 fluorides in compound 33 are sourced. Only two out of fourteen trityls in the production of the crystalline intermediate are accounted for. The implications of this should be explained experimentally or verbally. Apparently 12 arylations have already been performed before the formation of 33 is complete, which makes 33 as intermediate questionable. Does this not suggest that a majority of the reaction would not go via this cluster and that just the arylation starting from this cluster is the slowest which makes it isolable?

We agree that in the cluster many fluorides have been picked up already. This means that at the time when we isolate this intermediate the reaction is almost done. We do not think that the entire reaction proceeds through a single intermediate that looks exactly like compound 33. Instead, we believe that 33 shines light into the nature of potential reaction intermediates as a whole (they are likely less symmetric aggregates that do not crystallize). We have revised the discussion to express this more carefully. We now refer to 33 as a plausible intermediate and we added the following statement: "In fact, it is important to note that 12 arylations have already been performed before the formation of 33 is complete, i.e. the reaction is close to completion when the cluster is present at a concentration that is sufficient to allow its crystallization. It is therefore possible that a majority of the reaction proceeds through similar aggregates that are more reactive and cannot be isolated even under cryogenic conditions."

There is limited data present to support the same mechanism being active in primary fluorides. A concerted mechanism may become active where fluoride abstraction becomes challenging. This should be elaborated experimentally, computationally, or verbally. A possible experiment avoiding direct cation detection could be kinetic measurements in solvents of different polarity. If the ionic intermediate would be part of the rate determining path, polar solvents should significantly increase the reaction rate. A concerted mechanism should be less affected.

We thank this Reviewer for these suggestions which prompted a series of additional mechanistic experiments. We now add proof that both primary and secondary alkyl fluorides react through an ionic mechanism with intermediate carbocation-zincate ion pairs. In addition, we show more mechanistic evidence that the ion pair formation is followed by rapid C-C bond formation that outcompetes molecular reorientation between the carbocation and the zincate thus favoring retention of configuration. This is in agreement with the highly stereocontrolled C(sp³)-F bond functionalization of the cinchona alkaloid derivative 70 noted by Reviewer 1. We first conducted the suggested kinetic measurements in solvents of different polarity. Because the fluorophilicity of organozinc compounds is sensitive to Lewis-basic solvents we were limited in our selection. We chose hexafluorobenzene ($\epsilon = 2.05$) and trifluorotoluene ($\epsilon = 9.40$) to cover a significant difference in dielectric constants, ϵ . The arylation of 1-fluorooctane was monitored under otherwise identical conditions (see SI). We observed no discernable reactivity trend with respect to solvent polarity and both reaction showed almost identical kinetics. We believe this is in agreement with a mechanism based on short-lived carbocation-zincate ion pairs that are lipophilic and soluble in weakly polar solvents. The uncharged ion pairs react very quickly within the solvent shell (i.e. free ions do not need to be stabilized by solvent interactions) to the desired cross-coupling products which is apparent from the small amounts of rearrangement by-products formed via intrinsically fast intramolecular hydride shifts. Solvent stabilization effects are therefore negligible. We then decided to prove carbocation formation with a Friedel-Crafts trapping experiment using the primary alkyl fluoride 9 in 1,3-dimethoxybenzene as solvent. This was successful and shows that primary substrates also react through an ionic mechanism. We then performed the same experiment with the secondary fluoride 8 and again found evidence for carbocation formation (Figure 4C and SI). These results are further corroborated by the rearrangement studies with 8 and 9 that prove carbocation intermediate are formed but rapidly react as hydride shifts are outperformed (see Fig. 4D and SI). Finally, we obtained the enantioenriched secondary alkylfluorides 8 and 46 to provide further mechanistic insights into the C-C bond formation process. We observed considerable erosion of the ee's in the reactions with diphenylzinc but not full racemization. Comparison with literature reports revealed that the reaction of (S)-8 (92% ee) proceeds with retention of configuration and we isolated (S)-10 in 41% ee. The reaction with 46 (60% ee) gave 47 in respectable 33% ee. These results indicate that the aryl nucleophile delivery in the intermediate ion pair is fairly fast and able to compete with molecular tumbling processes that are likely to cause the erosion in the compound enantiopurities. All these new experiments point to a similar (stepwise) mechanism for primary, secondary and tertiary alkyl fluorides and are now discussed in detail in Fig. 4, the accompanying text and the SI.

Which intermediate zinc cluster is shown in fig. 5A? The bullet points do not seem to explain why the cluster is shown here. The bullet point claims that the cluster ion pair is crucial for a number of reasons which are likely but not proven. The CF bonds themselves are different to previously activated bonds. It seems if the cluster formation is the reason for the distinct

behaviour. Showing and discussing the cluster again in this figure is only repetition of previously displayed and discussed items and could be avoided to avoid confusion.

We agree that it is likely that the cluster 33 is crucial for several of the observed reaction features but isolation of similar clusters produced from primary or secondary alkyl fluorides is not possible as these compounds require higher reaction temperatures and intermediates are increasingly short-lived and less prone to form single crystals. We have therefore changed the discussion throughout the manuscript and also provide a more general manuscript title. We now don't refer to intermediate ion pair clusters when discussing the reaction with primary and secondary alkyl fluorides and also removed the cluster picture from Fig 5 to avoid confusion.

Reviewer #3 (Remarks to the Author):

The manuscript by Wolf describes a catalyst-free cross-coupling between alkyl fluorides and diaryl zinc species. Various alkyl fluorides are applied to the reaction, including primary, secondary, and tertiary alkyl fluorides. In particular, unactivated alkyl fluorides are good coupling partners for the current process, in sharp contrast to the previous reports that were limited to narrow substrate scope. Mechanistic studies reveal that short-lived ion-pairing organozincate clusters are responsible for the reaction, and a unique supramolecular reaction pathway enables rapid aryl transfer. This is a nice contribution to the C(sp²)-C(sp³) bond formation, which overcomes the limitations of transition-metal-catalyzed coupling, including β -hydride elimination and homodimerization. However, the substrate scope and functional group tolerance are not as good as the authors emphasized. Most substrates examined in the reaction are activated, such as propargyl and benzyl fluorides. The reaction tolerates aryl bromide, cyanide, and acetamide, but did not exhibit high functional group tolerance. Most importantly, all the alkyl fluorides are not readily available as the authors mentioned, which would regulate the synthetic utility of the approach. Overall, given the detailed mechanistic studies and the novel pathway involved in the reaction, the manuscript could be published in Nat. Commun. after revision.

We appreciate the support from this Reviewer. We show that the C-C bond formation is possible with unactivated primary, secondary and tertiary C(sp³)-F bonds as well as with benzylic, propargylic and acyl fluorides. The functional group tolerance is actually also very broad and includes nitrile, ketone, amide, ester, pyridine, furan, thiophene, lactam, lactone, nitro, carbonate, CF₂, CF₃, aryl halides including C(sp²)-F, alkene, alkyne, ether, and amino groups. In some cases, we have several functionalities present in one molecule. We now report that 1/3 of the alkyl fluorides used in this study are commercially available with 2/3 were easily prepared in one step using routine literature protocols.

1) The use of lithium salt to activate the C-F bond for functionalization is well-known. In one example Figure 2B, the addition of LiI to the reaction benefits the reaction efficiency (10); however, for substrate 9, poor yield was obtained by using LiI. Comments on these differences should be provided.

That is a very good observation. In fact, the reaction with 9 gave 80% octyl iodide which did not react further. This is now mentioned under the table in Figure 2B.

2) The preparation of alkyl fluorides should be clarified in the main text and SI. If the authors used a tedious procedure to prepare alkyl fluorides, it would decrease the synthetic utility of this approach.

We now provide information on commercially available and synthesized alkyl fluorides in the Methods section of this paper.

3) Typos

"(Fig. 2C and SI)" should be "(Fig. 1C and SI)."

Thanks, this has been corrected.

Thank you very much for the consideration of our work for publication in *Nature*

REVIEWERS' COMMENTS

Reviewer #1 (Remarks to the Author):

Although the reviewer remains uncomfortable with accepting this paper for Nature Communications for the reasons previously stated, the reviewer would like to respect the opinions of the editor and the other reviewers.

The reviewer has carefully read the responses of the authors to the comments. As a result, there are still some inadequacies and improvements would be appreciated.

In the table in Figure 2A, the authors respond that the conversion of compound 7 is 80%, but if that is the case, it should simply state "Conversion." In other words, "to 7" should be deleted. Similarly, "to 10" and "to 10 & 11" in the table in Figure 2B should also be deleted.

Reviewer #2 (Remarks to the Author):

The authors have thoroughly treated all suggestions and criticism that I voiced. My previous evaluation of the expected strong impact stands so that I am now convinced that this study should be published ASAP in Nature Commun.

Reviewer #3 (Remarks to the Author):

The authors have fulfilled the requests raised by the referees. The manuscript can be published in Nat. Commun.

Dear Reviewers:

We appreciate the very helpful comments and suggestions provided during the reviewing process. Our final comments are provided below.

Reviewer #1 (Remarks to the Author): Although the reviewer remains uncomfortable with accepting this paper for Nature Communications for the reasons previously stated, the reviewer would like to respect the opinions of the editor and the other reviewers. We regret this Reviewers' hesitation but do appreciate the very helpful comments provided previously. The reviewer has carefully read the responses of the authors to the comments. As a result, there are still some inadequacies and improvements would be appreciated. In the table in Figure 2A, the authors respond that the conversion of compound 7 is 80%, but if that is the case, it should simply state "Conversion." In other words, "to 7" should be deleted. Similarly, "to 10" and "to 10 & 11" in the table in Figure 2B should also be deleted. **The optimization studies focused on successful product formation and not simply C-F bond cleavage. We therefore believe it is more informative to list the conversion of the starting materials to the desired products rather than providing general starting material consumption data. This would not distinguish between conversion to the desired product and by-products and be less helpful.**

Reviewer #2 (Remarks to the Author): The authors have thoroughly treated all suggestions and criticism that I voiced. My previous evaluation of the expected strong impact stands so that I am now convinced that this study should be published ASAP in Nature Commun.

We appreciate the support of this Reviewer for our work.

Reviewer #3 (Remarks to the Author): The authors have fulfilled the requests raised by the referees. The manuscript can be published in Nat. Commun.

We appreciate the support of this Reviewer for our work.